# Debiasing Pretrained Text Encoders by Paying Attention to Paying Attention

## Abstract

Recent studies in fair Representation Learning have observed a strong inclination for Natural Language Processing (NLP) models to exhibit discriminatory stereotypes across gender, religion, race and many such social constructs. In comparison to the progress made in reducing bias from static word embeddings, fairness in sentence-level text encoders received little consideration despite their wider applicability in contemporary NLP tasks. In this paper, we propose a debiasing method for pre-trained text encoders that both reduces social stereotypes, and inflicts next to no semantic damage. Unlike previous studies that directly manipulate the embeddings, we suggest to dive deeper into the operation of these encoders, and pay more attention to the way they pay attention to different social groups. We find that most stereotypes are also encoded in the attention layer. Then, we work on model debiasing by redistributing the attention scores of a text encoder such that it forgets any *preference* to historically advantaged groups, and attends to all social classes with the same intensity. Our experiments confirm that we successfully reduce bias with little damage to semantic representation.

## 1 Introduction

Natural Language Processing (NLP) is increasingly penetrating real-world operations such as recruitment (Hansen et al., 2015), legal systems (Dale, 2019), healthcare (Velupillai et al., 2018) and Web Search (Nalisnick et al., 2016). Part of this success is attributed to the underlying embedding layer which encodes sophisticated semantic representations of language (Camacho-Collados & Pilehvar, 2018). The wide adoption of modern NLP models in critical domains has also inflicted a more thorough scrutiny. Recent research has uncovered some propensities of NLP models to replicate discriminatory social biases (Bolukbasi et al., 2016; Caliskan et al., 2017; May et al., 2019) which may cause unintended and undesired model behaviors with respect to social groups. Social bias in NLP is mainly caused by unbalanced mentions of attributes near advantaged groups in training data (Zhao et al., 2018a). For example, in most existing text corpora, very few cooks are referred to by male pronouns (e.g. he, him, himself) (Zhao et al., 2017). Accordingly, text encoders or language models trained on such data may use this shortcut to inadvertently disassociate cooks from men, and learn that cooking is a female attribute. Worse, NLP models may even amplify social biases if left unchecked (Bommasani et al., 2021).

Methods to *debias* static word embeddings such as Word2vec (Mikolov et al., 2013) or GloVe (Pennington et al., 2014) have been applied for gender, race and religion (Bolukbasi et al., 2016; Zhao et al., 2018b; Kaneko & Bollegala, 2019; Ravfogel et al., 2020). However, by the time NLP practitioners started casting more attention to the fairness problem of their models, they had already switched to the more powerful sentence-level transformers in the likes of BERT (Devlin et al., 2018), GPT3 (Brown et al., 2020) or T5 (Raffel et al., 2020) which owe their success to the novel self-attention mechanism (Vaswani et al., 2017). This leap in accuracy in several NLP tasks does not extend to fairness since research discovered social stereotypes in modern text encoders (May et al., 2019; Nadeem et al., 2020; Nangia et al.). To date, debiasing them remains comparatively under-explored.

Mitigating biases in text encoders is difficult for four reasons: (1) They are expensive to retrain, so conventional methods based on Counterfactual Data Augmentation (CDA) to rebalance group-attribute mentions (Zhao et al., 2018a; Webster et al., 2020) become prohibitive as they generate

more training data, and all debiasing attempts might be limited to either finetuning or adapting (Houlsby et al., 2019; Lauscher et al., 2021). (2) Static embeddings encode words whereas text encoders need context[1]. Thus, it is not straightforward to use existing debiasing techniques for static embeddings *off-the-shelf* as it is not clear how to generate context for single words. Previous work tackled this problem by either designing bleached sentence templates (May et al., 2019; Kurita et al., 2019) where they fill in the blanks with words of interest, or sampling sentences from large corpora where the words are mentioned (Liang et al., 2020a; Cheng et al., 2020), thus creating context. The former betrays the expressiveness of natural language while the latter suffers from sampling and pre-processing bias (Liang et al., 2020a). (3) The input space of text encoders is the set of all possible sentences, so we cannot debias every single input as it is done with static embeddings. (4) Text encoders are larger in capacity and complexity. This suggests that they can accommodate subtler and more sophisticated forms of stereotype, especially in their attention component, which renders bias imperceptible to existing detection methods as they are not designed to operate on attention.

Despite these difficulties, previous works (Liang et al., 2020a; Webster et al., 2020; Liang et al., 2020b; Cheng et al., 2020; Kaneko & Bollegala, 2021; Lauscher et al., 2021) addressed the problem of reducing bias from modern text encoders with different techniques. However, most of them make strong assumptions about the linearity of bias. Moreover, they operate on the embeddings produced by text encoders, and leave their most important block - attention - largely unrectified.

In this paper we explore attention-based debiasing. This approach stems from our observation that attention exhibits a great deal of social biases. We empirically show that this is the case, propose a novel method to reduce stereotypes from attention blocks, and demonstrate that it is effective in mitigating biases from sentence representations as a whole. Given an input sentence, our method compels the text encoder of interest to redistribute its internal attention scores such that each word in the input allocates the same attention for different social groups. Thus, it learns to forget previously encoded preferences, and generate fair representations, free of stereotypical influence. We also keep semantic information loss at a minimum while debiasing by distilling knowledge (Hinton et al., 2015; Gou et al., 2021) from an unaltered teacher text encoder. In this setting, we encourage the debiased model to copy the original attention from its teacher to minimize semantic offset. Unlike most previous work which focus only on gender, we address five bias types in our experiments (gender, race, religion, age and sexual orientation). We conduct likelihood- and inference-based evaluations to measure the intensity of bias in our final debiased models. Experiments demonstrate that the technique we propose effectively reduces bias, and outperforms existing debiasing methods.

## 2 RELATED WORK

In this section, we discuss related work about debiasing static word embeddings and sentence-level text encoders. Then, we shed some light on work done on the attention mechanism in general. It should be noted that bias at data level (Pryzant et al., 2020; Cryan et al., 2020) and in language generation tasks (Sheng et al., 2020; Sap et al., 2020; Dhamala et al., 2021) are also active and complementary areas of research. However, due to space limitations, they will not be discussed in this paper.

### 2.1 BIAS IN STATIC WORD EMBEDDINGS

The work of Bolukbasi et al. (2016) pioneered bias research in NLP by discovering that static word embeddings such as Word2Vec (Mikolov et al., 2013) or GloVe (Pennington et al., 2014) encode significant amounts of *binary* gender bias. They proposed *Hard-Debias*: a simple method to remove biases by projecting gender-neutral word embeddings onto a gender-free direction. Manzini et al. (2019) extended Hard-Debias to the multiclass setting where they also treat racial and religious stereotypes. In both works, the bias direction is defined by a manually pre-compiled list of stereotyped words. In contrast, Ravfogel et al. (2020) suggest a data-driven approach to learn bias directions with a linear classifier. Debiasing is then conducted by iteratively projecting word embeddings on the null space of the classifier's matrix. On the other hand, finetuning is the debiasing approach that attracted the widest adoption, either by using an autoencoder (Kaneko & Bollegala, 2019), attraction-repulsion mechanism (Kumar et al., 2020), or adversarial attacks (Xie et al., 2017;

---

[1]A word needs to be in a context (sentence or paragraph) in order to be correctly encoded

Li et al., 2018; Elazar & Goldberg, 2018). Unlike these post-processing methods, Zhao et al. (2018b) added a new fairness constraint to GloVe loss function, and retrained their fair word embeddings from scratch.

## 2.2 BIAS IN TEXT ENCODERS

Research on biases in sentence representations is dominated by detection rather than correction and mitigation. To date, there are three main approaches to detect stereotypes in text encoders: (1) **representation-based**: where vector relationships between different types of inputs are measured. For example, May et al. (2019) extended the WEAT test (Caliskan et al., 2017) into sentence vector space (SEAT), and compared the cosine similarity between representations of two sets of targets and two sets of attributes. All sentences in SEAT follow a predefined template. (2) **likelihood-based**: These approaches examine how often text encoders prefer stereotypes over anti-stereotypes. Preferences in this case are defined in terms of higher likelihoods as produced by language models using embeddings of the text encoders under study. Two benchmarks are widely used for measuring bias: StereoSet (Nadeem et al., 2020) and Crows-Pairs (Nangia et al.). Both datasets are organized in pairs or triples of minimally-distant sentences which differ only in the word(s) carrying a stereotypical connotation. (3) **inference-based**: These methods employ text encoders in downstream NLP tasks (Blodgett et al., 2020) such as natural language inference (Dev et al., 2020), sentiment analysis (Díaz et al., 2018) or language generation (Sap et al., 2020; Sheng et al., 2020). Bias in such settings is declared as the difference in outcome when the models are tested with the same input sentence, differing only in social groups.

Bias mitigation approaches are mostly inspired by debiasing static embeddings. In projection-based methods, Liang et al. (2020a) contextualize words into sentences by sampling them from existing corpora before applying Hard-Debias. Kaneko & Bollegala (2021) minimize the projection of sentence representations on a *learned* bias subspace, while Qian et al. (2019); Bordia & Bowman (2019); Liang et al. (2020b) add bias-reduction objectives to their loss functions. Another line of research uses CDA (Webster et al., 2020) to balance gender correlations in training data, while Lauscher et al. (2021) use adapters to reduce the large training time that CDA incurs. Finally, Cheng et al. (2020) use contrastive learning, and add a fair filter that minimizes mutual information between stereotypes and anti-stereotypes. In our work, rather than extending approaches from static embeddings, we focus on the self-attention mechanism which is characteristic of many text encoders, and show that fair attention leads to fair representations.

## 2.3 ATTENTION ANALYSIS IN TEXT ENCODERS

Clark et al. (2019) analyzed BERT's attention heads and found that some of them correspond remarkably well to linguistic patterns of coreference and syntax without additional training. Michel et al. (2019) observe that not all attention heads within a model are made equal. They also propose a pruning algorithm to reduce the energy footprint of these models by eliminating the least important heads without much attenuation to the overall performance. Given the convenient interpretabilty of attention, it has also been used in a myriad of visualization works (Vig, 2019; Hoover et al., 2020; Tenney et al., 2020; Bastings & Filippova, 2020) in an attempt to dissect and explain the inner functioning of text encoders. Most attention studies in text encoders are designed for analysis purposes. In contrast, we are the first to leverage the attention mechanism in order to make text encoders fairer and less stereotyped.

## 3 DEBIASING METHOD

### 3.1 MOTIVATING EXAMPLE & INTUITION

Despite the applicability of our work on any model that is built upon self-attention, we focus in this paper on models based on the encoder side of the transformer architecture, such as BERT (Devlin et al., 2018), RoBERTa (Liu et al., 2019), or ALBERT (Lan et al., 2019). This owes to the decoder side being usually used in auto-regression tasks, and less often to encode text. Transformers consist of multiple layers, each composed of a self-attention block followed by a feed-forward block to make embeddings. A self-attention block contains multiple heads. Each head transforms the input into

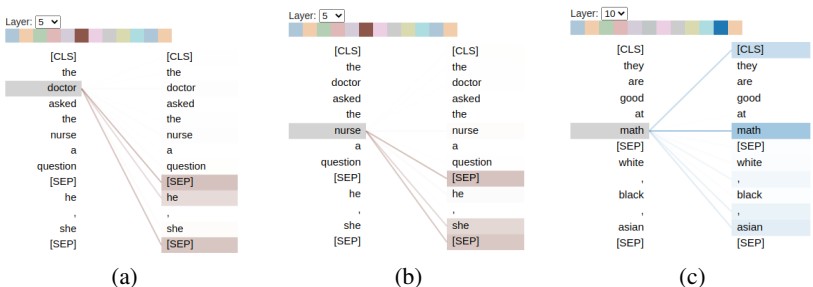

Figure 1: Attention patterns in BERT suggest the existence of potential gender and racial biases

attention weights between all pairs of tokens in the input sequence (Vaswani et al., 2017), such that each token learns to attend to its most related tokens, hence the prevalence of attention in defining the *understanding* of natural language. In this work, we assume that undesired social stereotypes are primarily encoded in the self-attention block. In order to verify this hypothesis, we show and analyze some attention maps of BERT in Figure 1[2]. Consider the following sentence *"The doctor asked the nurse a question."* Aiming to analyze how every word representation[3] relates to different social groups, we add a dummy second input consisting of words representing two distinct genders (**he** and **she** after the [SEP] token). Figure 1(a) illustrates that the token *doctor* pays much more attention to *he* than to *she*[4]. Figure 1(b) reveals that *nurse* attends to *she*. This finding suggests that gender stereotypes are deeply encoded in attention weights. Likewise, in Figure 1(c), the token *math* is more related to *asian* than to *white* or *black*, conforming to the famous racial stereotype casting asians as really good mathematicians. These examples align with our intuition that social stereotypes are first and foremost encoded in the self-attention block of text encoders before they propagate to their embeddings or predictions in downstream NLP tasks. Consequently, we propose *Att-D*, a finetuning method for reducing undesired biases of text encoders from their attention component.

The intuition of our debiasing strategy is as follows: Given that attention weights conform with undesired biases (e.g., *doctor* attending to **he**, and *nurse* to **she** in Figure 1), we aim to equalize the attention scores of every word in the input sentence with respect to social groups. Following the examples of Figure 1, Att-D redistributes the attention scores of *doctor* such that it attends to **he** and **she** with the same intensity, thus eliminating any preference toward one of the groups. However, alterations to the attention of *doctor* on the remaining words of the input sentence must be kept to a minimum in order not to corrupt the semantic understanding of the original text encoder.

## 3.2 DEBIASING WORKFLOW

Att-D consists of three steps: First, for each input sentence $s$ in the training corpus $\mathbb{S}$, we make an artificial second input $s_g$ consisting of words related to social groups (similar to the examples on Figure 1). Pretrained text encoders expect either one or two inputs before they produce attentions and embeddings. In this work, we use both $s$ (as first input) and $s_g$ (as second input) separated by [SEP] token. Consequently, the resulting self-attention includes both $s$ and $s_g$. The second step of Att-D equalizes the attention weights of all heads in all the layers of the text encoder of interest such that each token in $s$ pays the same amount of attention to tokens of $s_g$, thus eliminating preferences and stereotypes. Finally, we minimize semantic loss by compelling our model to learn the original semantics from an *unaltered* teacher model by copying its internal attention in a knowledge distillation setting (Hinton et al., 2015).

We schematize the operation of Att-D in Figure 2. $Gr_1$, $Gr_2$ and $Gr_n$ in the figure correspond to the tokens of $s_g$. Both matrices represent one attention head of the text encoder before (left) and after (right) debiasing. The matrices should be read in rows. Each row depicts the attention weights of the corresponding token on all the other tokens of the input $(s + s_g)$[5].

---

[2]The figures are produced using bertviz tool: https://github.com/jessevig/bertviz

[3]According to BERT in this example

[4]Dark colors correspond to high attention scores, and light colors indicate low attention scores

[5][CLS] token (vector representation of $s$) is also included for attention calibration. Details in Appendix

The matrices are conceptually split in four blocks: (1) attentions of $s$ on $s$, (2) attentions of $s$ on $s_g$, (3) attentions of $s_g$ on $s$, and (4) attentions of $s_g$ on $s_g$. Debiasing consists in making the columns of block 2 equal. In other words, each token in $s$ pays the same amount of attention to all the groups as indicated in the right side of Figure 2. Ideally, debiasing should also preserve the semantics of the original text encoder. That is why block 1 of Figure 2 should be kept unchanged. Both blocks 3 and 4 are irrelevant to the results, since they denote attentions of our artificially inserted second input $s_g$. Besides, they do not participate in defining neither fairness nor representativeness of the text

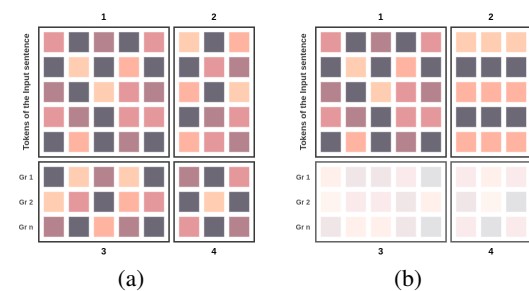

Figure 2: Overview of an attention head before (a) and after (b) debiasing

encoder. So, we do not impose any restrictions on them. In the following, we describe each step of Att-D in detail.

### 3.2.1 GENERATING AUGMENTED INPUTS

The first step involves identifying bias types that we want to mitigate from pretrained text encoders, such as gender, race, religion. This is achieved by defining a set of tuples $\mathbb{G}$ such that $\mathbb{G} = \{\mathcal{T}_1, \mathcal{T}_2, ..., \mathcal{T}_k\}$ where each $\mathcal{T}_i$ describes social groups of a given bias type, or their attributes. In the case of binary gender, $\mathcal{T}_i = \{"male", "female"\}$ or $\mathcal{T}_i = \{"he", "she"\}$ are both possible definitions (and many others are possible for non-binary cases). Table 1 shows some

Table 1: Examples of group tuples per bias

| binary gender | age |
|---|---|
| man, woman | old, young |
| he, she | elderly, youth |

tuples that we use in Att-D. Unlike previous work (Liang et al., 2020a; Kaneko & Bollegala, 2021), we do not aim to learn bias subspaces. Therefore, Att-D charges next to no constraint in the definition of $\mathbb{G}$, contrary to existing methods which need group definitions to be formulated in a certain way, and given in enough quantities. On the other hand, providing a single tuple per bias type is sufficient for Att-D to function, even though more contributes to better precision.

During debiasing, we pick a tuple $\mathcal{T}_i$ from $\mathbb{G}$ randomly and construct $s_g$, a bleached sentence formed by words of $\mathcal{T}_i$. For example, given Table 1, $s_g$ can be *"man, woman"* or *"old, young"*. The input to the text encoder is both the original input sentence $s$ and the artificial one $s_g$. Pretrained text encoders separate the two halves of the input with a special token [SEP] as illustrated in Figure 1, and compute attention maps for the entire sequence ($s + s_g$).

### 3.2.2 EQUALIZING ATTENTIONS ON SOCIAL GROUPS

After obtaining attention maps of the augmented input from Section 3.2.1, which is produced by the pretrained text encoder of interest $E$, we make each token of $s$ pay equal amounts of attention to the tokens of $s_g$ which define social groups. The rationale is to eliminate any inclination for $E$ to prefer a social group to the detriment of others. Suppose $\mathbf{A}^{l,h,s,s_g} = Attn(s, s_g; l, h)$ is the attention matrix at layer $l$, head $h$ of the encoder $E$, computed from the input $s + s_g$. Here, we make the reasonable assumption that $s_g$ contains at least two social groups.[6] In this spirit, equalizing attention vectors of block 2 (as defined in Figure 2) is equivalent to making them equal to a pivot vector. In our method, we consider the attention vector of $s$ on the first social group as the pivot (first column in block 2 of Figure 2), and minimize the mean square error between the pivot and the attention vectors of $s$ on the other groups, one at a time. The equalization loss is given by Equation 1.

$$L_{equ} = \sum_{s \in \mathbb{S}} \sum_{l=1}^{L} \sum_{h=1}^{H} \sum_{i=2}^{||s_g||} ||\mathbf{A}^{l,h,s,s_g}_{:\sigma,\sigma+1} - \mathbf{A}^{l,h,s,s_g}_{:\sigma,\sigma+i}||_2^2 \qquad (1)$$

---

[6]Biases are usually about making one or more groups (dis)advantaged with respect to the others, hence the existence of at least two groups per bias type

where $L$ is the number of layers of the text encoder, $H$ the number of heads, $||s_g||$ the number of social groups in $s_g$ and $\sigma$ is the position of the special token [SEP] that marks the end of $s$ and the beginning of $s_g$. As can be seen, $\mathbf{A}_{:\sigma,\sigma+1}^{l,h,s,s_g}$ is the pivot vector containing attention scores of $s$ on the first social group token (whose position is directly after [SEP], i.e., $\sigma+1$). Equation 1 forces attention scores on subsequent social groups to be the same as on the first one, thus making them all equal.

### 3.2.3 PRESERVING SEMANTIC INFORMATION

Text encoders must preserve their ability to represent natural language and keep the same performance on downstream NLP tasks. For this reason, debiasing must ensure that useful information is preserved as much as possible. We minimize semantic information loss in a knowledge distillation setting (Hinton et al., 2015; Gou et al., 2021). We cast the text encoder that we want to debias as the *student* model, and recruit another model to play the role of the *teacher*. We do not apply our debiasing strategy on the teacher since it provides a reference to the original unaltered language representations. We distill semantic information in the form of attention maps from the teacher and instill it in the student. Stated differently, we compel the student to learn from the teacher and reproduce its attention scores for every input sentence in the training corpus $\mathbb{S}$. In this work, the student and the teacher models *must* share the exact same architecture in order to make knowledge distillation flow correctly from source to target, unlike other works in knowledge distillation (Gou et al., 2021) wherein the student is much smaller and should be more compact than the teacher for model compression purposes.

As in Section 3.2.2, let $\mathbf{A}^{l,h,s,s_g}$ be the attention of the student model at layer $l$, head $h$ with $s$ and $s_g$ as input. Likewise, let $\mathbf{O}^{l,h,s,s_g}$ define the same attention matrix, but for the original teacher model. We formalize the preservation of semantic information as a regularizer where we minimize the squared $l_2$ distance between the student's and the teacher's attention scores according to the following:

$$L_{distil} = \sum_{s \in \mathbb{S}} \sum_{l=1}^{L} \sum_{h=1}^{H} ||\mathbf{A}_{:\sigma,:\sigma}^{l,h,s,s_g} - \mathbf{O}_{:\sigma,:\sigma}^{l,h,s,s_g}||_2^2 \tag{2}$$

where $L$ is the number of layers, $H$ is the number of heads, and $\sigma$ is the position of the [SEP] token. As can be seen from Equation 2, the student learns only to replicate block 1 (as in Figure 2) of the attention matrices. This is because block 1 contains attention scores of the original input sentence $s$ on itself, thus encoding an important aspect of semantics. We force the student not to reproduce the attention distribution on social groups (block 2) from the teacher since these are fundamentally biased, and are left to the care of our debiasing objective. We describe the overall training objective as a linear combination of the previously defined losses, with $\lambda$ as a hyperparameter to control the weight of debiasing over semantic preservation.

$$Loss = L_{distil} + \lambda L_{equ} \tag{3}$$

### 3.3 NEGATIVE SAMPLING & LAYER SELECTION

The strict application of Att-D as discussed so far may accidentally lead to some undesired spurious phenomena. While learning to equalize attention on social groups that constitute the second half of the input, the text encoder might potentially bear the risk of distributing its attention uniformly on *any* second half of the input, no matter what it is. This is particularly alarming when the text encoder is subsequently employed in double-sentence tasks (Wang et al., 2018) such as semantic textual similarity, paraphrase detection or sentence entailment.

To overcome the above obstacle, we introduce negative sampling. Instead of using words related to social groups in order to generate the artificial second input $s_g$, we randomly sample words (negative examples) from the vocabulary. In this case, we do not equalize the attentions but compel the student to copy its teacher even for blocks 2, 3 and 4. We do this in order to prevent the text encoder from learning to assign the same attention weight to all tokens of the second input when these do not define social groups. We control the ratio of negative examples with a hyperparameter $\eta$.

Another concern when debiasing text encoders is that their layers do not necessarily encode the same information. Bhardwaj et al. (2021) found that BERT layers display widely different reactions when probed with a gender-detection classifier. This means that they do not encode gender stereotype identically. Therefore, it is not clear which layers and/or attention heads are best for debiasing. To investigate this issue, we consider seven settings: debiasing *all* layers, *first 6*, *first 3*, *last 6*, *last 3*, and alternating layers with strides of *one*[7] or *two*[8]. We find that working with all layers yields the best results. So, unless otherwise specified in this paper, debiasing concerns all the layers.

# 4 EVALUATION

In this section, we first describe our experimental setup, then evaluate Att-D from two viewpoints: *fairness* and *representativeness*. Unlike previous work in debiasing text encoders (Liang et al., 2020a; Cheng et al., 2020; Kaneko & Bollegala, 2021), we do not conduct representation-based evaluations - SEAT (May et al., 2019) and WEAT (Caliskan et al., 2017) - in this paper to measure bias since their effectiveness for such a purpose has been questioned (Silva et al., 2021). On the other hand, we evaluate fairness with both likelihood- and inference-based methods. Other quantitative and qualitative evaluations can be found in the appendix[9].

## 4.1 DEBIASING SETUP

In the experimental evaluation we focus on five different bias types which are commonly addressed by the bias literature and debiasing approaches. To facilitate comparison, we also follow existing literature in defining the social groups for each type of bias, although the approach presented here is not restricted to that, and can be leveraged for both other kinds of biases and for a more inclusive definition of the groups. In the experiments we show results of debiasing based on **(binary) gender** (*male*, *female*), **race** (*white*, *black*, *asian*, *hispanic*), **religion** (*muslim*, *christian*, *jewish*, *buddhist*), **age** (*young*, *old*) and **sexual orientation** (*heterosexual*, *non-heterosexual*). We leverage the definition words from previous work (Liang et al., 2020a) and augment them with our own tuples. The full list can be found in the appendix. We apply our debiasing method on BERT[10] (Devlin et al., 2018), and use the News-commentary-v15 corpus[11] as training data. It contains 223,153 sentences of which we use 80% for training and 20% for development.

## 4.2 EVALUATIONS OF FAIRNESS

### 4.2.1 LIKELIHOOD-BASED EVALUATION

We use the publicly available subsets of two stereotype benchmarks: StereoSet (Nadeem et al., 2020) and Crows-Pairs (Nangia et al.). Both provide likelihood-based diagnostics to measure how often stereotypes are considered more likely than anti-stereotypes by the text encoder of interest[12]. Crows-Pairs is organized in pairs of minimally distant sentences, i.e. sentences that differ only in one word, usually the one describing a social group. We pick the following example from the dataset: ***Jews are such greedy people*** (Stereotype) versus ***Christians*** *are such greedy people* (Anti-stereotype). Crows-Pairs uses word likelihoods produced by the text encoder to measure how often stereotypes are ranked higher than anti-stereotypes. An ideal *unbiased* text encoder should score 50% in the Crows-Pairs challenge, meaning that it prefers neither stereotypes nor anti-stereotypes. 100% and 0% are both undesired extreme outcomes. In contrast, StereoSet adds a third absurd sentence to capture the language modeling capabilities of the text encoder in addition to measuring bias. The triple of sentences in StereoSet are also minimally distant but differ in the attribute word, not in the group. For example, in *Girls tend to be more [MASK] than boys*, StereoSet generates the evaluation sentences by replacing the mask with the following attributes: ***soft*** to denote stereotype, ***determined*** for anti-stereotyping, and ***fish*** to make an unrelated sentence. As in Crows-Pairs, StereoSet quantifies how

---

[7]Layers 2, 4, 6, 8, 10 and 12 of BERT

[8]Layers 4, 8 and 12 of BERT

[9]We provide code and data as supplementary material to this submission

[10]In the appendix, we also apply Att-D on ALBERT, RoBERTa, DistilBERT and SqueezeBERT

[11]http://www.statmt.org/wmt20/translation-task.html

[12]The text encoder must first be fine-tuned on a language modeling objective

Table 2: Language modeling (lm) and Stereotype scores (ss) of different text encoders on StereoSet

| Models | Original BERT | | Sent-D | | Kaneko | | Att-D⁻ | | Att-D | |
|---|---|---|---|---|---|---|---|---|---|---|
| Overall (lm/ss) | 83.70 | 56.04 | 81.39 | 54.71 | 85.58 | 56.04 | 80.92 | 53.37 | 83.34 | **53.04** |
| gender (lm/ss) | 82.35 | 62.75 | 76.67 | 53.33 | 83.73 | 58.82 | 73.33 | **51.37** | 78.24 | 53.73 |
| race (lm/ss) | 86.28 | 54.68 | 85.40 | 55.09 | 87.47 | 56.24 | 85.03 | 54.37 | 86.28 | **51.87** |
| religion (lm/ss) | 87.82 | 56.41 | 88.46 | **51.28** | 85.90 | 57.69 | 85.90 | 55.13 | 88.46 | 53.85 |
| profession (lm/ss) | 80.66 | 55.50 | 77.44 | 55.01 | 83.87 | 54.76 | 77.94 | **52.66** | 80.96 | 54.14 |

much the model prefers stereotypes over anti-stereotypes. Moreover, it also measures how often meaningful sentences rank higher than unrelated sentences in order to assess the quality of the text encoder in terms of representativeness. An ideal model should have a stereotype score (*ss*) of 50% and a language modeling (*lm*) of 100%.

We compare our method against the original BERT base model to see the effect of debiasing[13]. Also, for fair comparisons against previous work, we decided to include the baselines whose final debiased models have been published in order to avoid errors of training and/or tuning hyperparameters. Thus, we compare Att-D against Sent-D (Liang et al., 2020a) and the debiasing procedure proposed by Kaneko & Bollegala (2021). Sent-D currently holds state-of-the-art performance in debiasing. We also conduct a simple ablation study by training without negative examples (*Att-D⁻*). We finetune Att-D and the baselines on language modeling to be able to conduct likelihood-based experiments. Tables 2 and 3 report the evaluation results on StereoSet and Crows-Pairs respectively.

We see that the original BERT contains significant levels of biases (56.04 in StereoSet and 60.48 in Crows-Pairs). It is important to note that Kaneko & Bollegala (2021) only focused on gender bias, which is clear in Table 2 where only gender stereotype has been reduced. We observe that focusing on one bias type can make text representations even more biased for the other dimensions as can be seen in Kaneko for race and religion (Table 2), and for sexual orientation and disability (Table 3). In contrast, Att-D always reduces the intensity of stereotyping in BERT (up to 9.53%), and yields the best

Table 3: Bias measurements of different text encoders on Crows-Pairs

| Models | BERT | Sent-D | Kaneko | Att-D⁻ | Att-D |
|---|---|---|---|---|---|
| Overall | 60.48 | 56.90 | 57.82 | 57.23 | **55.7** |
| gender | 58.02 | 51.53 | 57.63 | 53.05 | 57.36 |
| race | 58.14 | 55.23 | 53.68 | 53.68 | 51.15 |
| religion | 71.43 | 60.0 | 64.76 | 69.52 | 64.76 |
| age | 55.17 | 51.72 | 54.02 | 54.02 | 43.68 |
| sexual orientation | 67.86 | 70.24 | 69.05 | 66.67 | 58.33 |
| nationality | 62.89 | 56.6 | 59.12 | 61.01 | 57.86 |
| disability | 61.67 | 65.0 | 68.33 | 63.33 | 60.0 |

results overall[14]. We also notice that we manage to reduce biases linked to dimensions we did not include in our design such as profession, nationality and disability. We speculate that these bias types are connected to those we worked on mitigating. Therefore, we conjecture that reducing multiple biases at the same time meets better success in mitigating unforeseen stereotypes than working on every bias type separately.

### 4.2.2 INFERENCE-BASED EVALUATION

This approach of measuring bias builds on the intuition of Dev et al. (2020) stating that biased representations lead to invalid inferences, whose ratio quantifies bias. They construct a challenge benchmark for the natural language inference task where every hypothesis should be *neutral* to its premise. For example, suppose that the premise is *The **driver** owns a van* and the hypothesis is *The **man** owns a van*. The hypothesis neither entails nor contradicts the premise. If the predictions of a classifier deviate from neutrality, the underlying text encoder is doomed as biased. All the data points in Dev et al. (2020) follow the same structure of the example above, and span across gender, religion and race. Suppose that the set contains $M$ instances, and let the predictor's probabilities of the $i^{th}$ instance for entail, contradict and neutral be $e_i$, $c_i$ and $n_i$. Following Dev et al. (2020), we report three measures of inference-based bias: (1) Net Neutral (**NN**): $NN = \frac{1}{M} \sum_{i=1}^{M} n_i$; (2) Fraction Neutral (**FN**): $FN = \frac{1}{M} \sum_{i=1}^{M} \mathbf{1}_{n_i=max(e_i,c_i,n_i)}$; (3) Threshold $\tau$ (**T:**$\tau$): $T : \tau = \frac{1}{M} \sum_{i=1}^{M} \mathbf{1}_{n_i>\tau}$.

---

[13]Results of BERT large, ALBERT, RoBERTa, DistilBERT and SqueezeBERT are in the appendix

[14]The closer the stereotype score is to 50%, the better

Table 5: Performance of different models on GLUE tasks. The table shows *accuracy* scores for **sst2**, **rte**, **wnli**, and **mnli** for both matched and mismatched instances; *f1* for **mrpc**; *spearman correlation* for **stsb**; and *matthews correlation* for **cola**

| Models | Single sentence tasks | | | | Double sentence tasks | | | | | | | | | | | |
|---|---|---|---|---|---|---|---|---|---|---|---|---|---|---|---|---|
| | sst2 | | cola | | stsb | | mrpc | | mnli (m) | | mnli (mm) | | rte | | wnli | |
| BERT | 92.78 | - | 56.05 | - | 88.97 | - | 92.25 | - | 83.54 | - | 82.68 | - | 70.04 | - | 45.07 | - |
| Sent-D | 91.63 | -1.15 | **59.08** | **+3.03** | 89.58 | +0.61 | 90.12 | -2.13 | **84.97** | **+1.43** | 83.51 | +0.83 | 68.95 | -1.09 | 28.17 | -16.9 |
| Kaneko | 91.97 | -0.81 | 56.50 | +0.55 | 88.44 | -0.53 | 90.69 | -1.56 | 84.48 | +0.94 | 83.66 | +0.98 | 59.93 | -10.11 | 52.11 | +7.04 |
| Att-D | **92.66** | **-0.12** | 55.22 | -0.83 | **89.62** | **+0.65** | **91.22** | **-1.03** | 84.63 | +1.09 | **84.19** | **+1.51** | **70.40** | **+0.36** | **53.52** | **+8.45** |

In this experiment, we finetune text encoders of interest with the **MNLI** dataset for natural language inference (Wang et al., 2018). An ideal bias-free model should score 1 in all three measures. We report our findings in Table 4 after transforming them into percentages. Our method outperforms the original model and the baselines. This result shows that Att-D not only reduces bias at surface-level as reflected by likelihood-inspired evaluation methods, but also mitigates stereotypes in real world inference settings, unlike Sent-D which produces positive results with Crows-Pairs but comes short of meeting the same success in this experiment. In the next section, we show that these findings are meaningful since the entailment accuracy is not hurt after debiasing.

Table 4: Inference-based bias measurements. Best scores are highlighted with **bold character**, underlined, or marked with † for **gender**, race and religion† respectively

| Model | Bias type | NN | FN | $\tau$:0.5 | $\tau$:0.7 |
|---|---|---|---|---|---|
| BERT | gender | 00.59 | 00.16 | 00.15 | 00.12 |
| | race | 75.96 | 76.57 | 76.51 | 74.91 |
| | religion | 43.47 | 43.55 | 43.45 | 41.77 |
| Sent-D | gender | 00.94 | 00.38 | 00.33 | **00.24** |
| | race | 59.61 | 59.28 | 59.20 | 56.22 |
| | religion | 29.64 | 29.08 | 29.02 | 27.24 |
| Kaneko | gender | 00.57 | 00.14 | 00.12 | 00.08 |
| | race | 84.24 | 84.84 | 84.80 | 83.26 |
| | religion | 69.27† | 69.80† | 69.72† | 67.66† |
| Att-D | gender | **01.31** | **00.43** | **00.35** | 00.21 |
| | race | 93.31 | 93.94 | 93.90 | 93.04 |
| | religion | 68.51 | 69.08 | 68.95 | 66.97 |

### 4.3 Evaluations of Representativeness

We use GLUE benchmark (Wang et al., 2018) to verify whether the debiased text encoder still holds enough semantic information to be applicable in downstream NLP tasks. In essence, GLUE assesses the natural language understanding capabilities of NLP models. So, it constitutes a suitable stack to evaluate the semantic preservation of Att-D. In this experiment, we finetune BERT on seven different tasks from GLUE and show the results in Table 5. We also report the difference in accuracy between original BERT and each of the debiasing baselines. Surprisingly, Att-D not only preserves semantic information, but enhances it in most GLUE tasks as reflected in an increase in accuracy from BERT.

## 5 Conclusion and Future Work

In this paper, we proposed a finetuning approach to debiasing that trains the text encoder to distribute its attention equally on different social groups. Experiments demonstrate that bias is successfully reduced without harm to semantic representativeness. However, we are aware of the following limitations: (1) our definitions of biases are simplified. There are more social divisions in the real world than the five dimensions we studied. Besides, bias types can be correlated in intricate ways such as the links between *race*, *nationality* and *ethnicity*. Moreover, it is not clear which or how many groups to include. For these reasons, we follow previous work and constrain our study to simple definitions targeting the most widely spread groups. We plan to study the effect that the choice of definition tuples and their order impose overall. (2) We equalize the attention scores of every word in the input. However, some words are inherently charged with a strong inclination toward one group, e.g., *beard* to **male** or *pregnant* to **female**. Such words need not be debiased, which requires compiling expensive lists of related words for every social group and protecting them from attention equalization. In this work, we rely on knowledge distillation to retain as much useful semantic information as possible. (3) Current bias detection experiments have positive predictive ability, which means that they can only detect the presence of bias, not the absence of it. Although contemporary evaluation tools demonstrate that our method successfully reduced social biases from sentence representations, it is possible that bias is still lurking in shapes and forms that our experimental lenses failed to detect. We plan to address these limitations in future work.

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

# A APPENDIX

## A.1 TRAINING HYPERPARAMETERS

We used Adam optimizer (Kingma & Ba, 2014) with a learning rate of $5e^{-6}$ for 3 epochs. We keep the betas to their default values (0.9, 0.999) as in PyTorch implementation (Paszke et al., 2017). We set the loss coefficient $\lambda$ to 2.0 and the negative ratio $\eta$ to 0.8 meaning that in 80% of the iterations, we use negative examples whose number we set to 5 in each negative iteration. We only finetuned the values of $\lambda$, $\eta$, the learning rate, and the number of epochs. We conducted the hyperparameter search manually on the development set.

As for GLUE experiments, we follow the experimental setup of Devlin et al. (2018) and train each task for 3 epochs with a learning rate of $2e^{-5}$ on their respective training data.

## A.2 SOCIAL GROUPS USED IN THE EXPERIMENTS

While the approach is independent of the definition of social groups and categories (it could work for any kind of grouping, e.g., cuisine styles or sports), in the experiment we focus on groups commonly used in the debiasing literature: binary gender, religion, race, age and sexual orientation. This is to facilitate comparison, but nothing in the approach prevent it from being used with broader and more inclusive groups. This being said, we have not experimented yet with debiasing where a dimension is divided in dozens of categories.

We list the definition tuples that we used in Table 6. We show that Att-D does not incur strict rules for defining social groups, unlike previous work (Bolukbasi et al., 2016; Kaneko & Bollegala, 2019; 2021) that require the definition words to be organized in a predefined format (pairs of words or bag of words for every group), and provided in relatively large quantities. We can see from Table 6 that it is sufficient to define one tuple per bias type (e.g., race) if the tuples are hard to come by. Also, the tuples need not be of the same size (e.g., in religion there is a missing word for *buddhist* group since it is not clear which word to use in that tuple). This desired property owes to the fact that Att-D does not learn subspaces or directions for every bias type as previous works do (Bolukbasi et al., 2016; Kaneko & Bollegala, 2019; Kumar et al., 2020; Kaneko & Bollegala, 2021). In contrast, Att-D uses the tuples in order to equalize the attentions of the input sentence, and make the words therein attend to the groups with the same intensity. These example categories used in experiments are neither complete nor exhaustive, and in some experiments also include terms possibly considered inappropriate but that appear in the corpus and we may still want to debias from (such as using "straight" to define heterosexual).

## A.3 CHOICE OF LAYERS

Transformer-based text encoders consist of many layers. It is not clear which layers to choose for debiasing since bias information is spread out across all of them. In this experiment, we try different

Table 6: Full list of definition tuples for bias types and social groups used in this work

| gender | | religion | | | |
|---|---|---|---|---|---|
| *male* | *female* | *muslim* | *christian* | *jewish* | *buddhist* |
| man | woman | muslim | christian | jewish | buddhist |
| boy | girl | muslims | christians | jews | buddhists |
| father | mother | islam | christianity | judaism | buddhism |
| brother | sister | mosque | church | synagogue | temple |
| grandfather | grandmother | quran | bible | torah | |
| son | daughter | imam | priest | rabbi | monk |
| gentleman | lady | mohammad | jesus | moses | buddha |
| he | she | | | | |
| his | her | | | | |
| himself | herself | | | | |

| race | | | | age | | sexual orientation | |
|---|---|---|---|---|---|---|---|
| *white* | *black* | *asian* | *hispanic* | *old* | *young* | *heterosexual* | *non-heterosexual* |
| white | black | asian | hispanic | old | young | straight | gay |
| | | | | elderly | youth | straight | lesbian |
| | | | | adult | child | heterosexual | homosexual |
| | | | | senior | junior | heterosexual | bisexual |
| | | | | adult | teenager | | |

Table 7: Language modeling (lm) and Stereotype scores (ss) of different layer combinations on StereoSet. Underlined depicts the best language modeling score, while **bold** shows the best stereotype score

| Models | first 3 | | first 6 | | last 3 | | last 6 | | 1-stride | | 2-stride | | all | |
|---|---|---|---|---|---|---|---|---|---|---|---|---|---|---|
| Overall (lm/ss) | 83.17 | 54.28 | 78.80 | 54.04 | 82.51 | 54.13 | 81.92 | 54.33 | 82.70 | 54.42 | 82.68 | 54.04 | 83.34 | **53.04** |
| gender (lm/ss) | 78.04 | 55.29 | 71.96 | 55.69 | 78.43 | 56.08 | 77.65 | 55.69 | 76.47 | 55.29 | 78.43 | 54.51 | 78.24 | **53.73** |
| race (lm/ss) | 87.11 | 54.05 | 83.16 | 54.16 | 85.71 | 54.05 | 85.40 | 54.57 | 85.76 | 54.26 | 86.38 | 52.91 | 86.28 | **51.87** |
| religion (lm/ss) | 87.82 | **51.28** | 87.82 | 57.69 | 85.90 | 52.56 | 88.46 | 57.69 | 88.46 | 53.85 | 86.54 | 60.26 | 88.46 | 53.85 |
| profession (lm/ss) | 79.67 | 54.51 | 74.91 | **53.03** | 79.67 | 53.77 | 78.49 | 53.28 | 80.47 | 54.39 | 79.23 | 54.64 | 80.96 | 54.14 |

debiasing settings in which we select different layer combinations of BERT to work on: ***all*** layers, ***first 6***, ***first 3***, ***last 6***, ***last 3***, and alternating layers with strides of ***1*** (layers 2, 4, 6, 8, 10 and 12) or ***2*** (layers 4, 8 and 12). We apply the debiasing method proposed in this paper, and report both language modeling and stereotype scores of StereoSet benchmark in Table 7.

The results show that it is safest to equalize attention heads of all layers of the text encoder under study, since it produces the best scores both in terms of language modeling and stereotype. Our findings go in tandem with those of Liang et al. (2020b); Kaneko & Bollegala (2021); Bhardwaj et al. (2021) who found that reducing bias from all layers usually is the best option.

## A.4 EFFECT OF NEGATIVE EXAMPLES ON REPRESENTATIVENESS

We remind that the introduction of negative examples to training serves in forcing the text encoder not to rely on a dangerous shortcut which is distributing its attention uniformly on all the tokens constituting the second half of the input, no matter what the input is. This is particularly important in double-sentence tasks where the text encoder is given two input sentences. In addition to Tables 2 and 3 which highlighted the effect of negative sampling on the final stereotype scores, the primary goal of using negative examples remains the preservation of the text encoder's representativeness. In Table 8, we report the performance of *Att-D* and *Att-D⁻* with and without negative examples respectively on GLUE tasks. Unsurprisingly, the lack of negative examples does not damage the performance of single-sentence tasks since these ignore the second half of the input altogether. However, in double-sentence tasks where both halves are used for prediction, Table 8 shows that negative sampling plays a pivotal role in preserving the semantics of text encoders, and bypassing the side effects inflicted by attention equalization.

Table 8: Effect of negative examples on GLUE tasks. The table shows *accuracy* scores for **sst2**, **rte**, **wnli**, and **mnli** for both matched and mismatched instances; *f1* for **mrpc**; *spearman correlation* for **stsb**; and *matthews correlation* for **cola**

| Models | Single sentence tasks | | Double sentence tasks | | | | |
|---|---|---|---|---|---|---|---|
| | sst2 | cola | stsb | mrpc | mnli (m/mm) | rte | wnli |
| BERT | **92.78** | 56.05 | 88.97 | **92.25** | 83.54 / 82.68 | 70.04 | 45.07 |
| Att-D | 92.66 | 55.22 | **89.62** | 91.22 | **84.63 / 84.19** | **70.40** | **53.52** |
| Att-D⁻ | 92.32 | **56.25** | 89.12 | 80.44 | 84.59 / 83.96 | 58.12 | 39.44 |

## A.5 VISUALIZING DEBIASING RESULTS

In this experiment, we aim to visualize the effects of debiasing on attention weights. We only focus on binary gender bias for two reasons: First, it is easier to visualize binary variables on a 2D plane than multiclass variables (such as race, religion...). Second, gender is the most well studied bias type (Bolukbasi et al., 2016; Caliskan et al., 2017; May et al., 2019), so linguistic resources and vocabularies for gender exist and are well documented. We use the vocabulary words compiled by Kaneko & Bollegala (2019) and categorized into three non-overlapping subsets: (1) **Male-definition** $\Omega^M$ whose corresponding words are exclusively male-gendered such as *father*, *king* or *uncle*. (2) **Female-definition** $\Omega^F$ which is a set of inherently female words (*mother*, *queen*, *aunt*...). (3) **Gender-stereotype** $\Omega^S$ which is constituted of words that are not gendered by definition, but that carry a strong gender stereotype such as *doctor* being attributed to **male** or *nurse* to **female**.

For every word $w \in \Omega^M \cup \Omega^F \cup \Omega^S$, we extract sentences from the News-commentary-v15 corpus where $w$ is mentioned. We denote this set as $S^w$. Then, for every sentence $s \in S^w$, we append the dummy input *"man, woman"* as explained in Sections **??** and 3.2.1. The augmented input $s'$ is then fed to the text encoder of interest (BERT base in this experiment), and we collect the attention scores of $w$ on the second-half tokens **man** and **woman**. Finally, for every word $w \in \Omega^M \cup \Omega^F \cup \Omega^S$, we take the mean of its attention scores in $S^w$. By the end of this procedure, we have for every word $w$ its attention score on the words **man** ($a_m^w$) and **woman** ($a_f^w$) as computed on the News-commentary-v15 corpus which includes overall 223,153 sentences. We take the difference $a_m^w - a_f^w$ which indicates the preference of the text encoder to consider $w$ as male (positive difference) or female (negative difference). The absence of gender bias is reflected in difference scores near zero.

We plot the results in Figure 3 where the x-axis represents the differences $a_m^w - a_f^w$, and the y-axis random values to separate the words vertically. Stereotype words (green dots) should have values near 0, which is not the case in Figure 3(b). This means that BERT has a strong preference for one of the genders, and is thus heavily biased. In contrast, our method brings the attention of stereotype words near 0, meaning that they prefer neither male nor female connotations. Moreover, the spread of stereotype words in Figure 3(d) is narrower than male- or female-oriented words, which is desired since these are inherently gendered and must pick a side. This result strengthens the claim that Att-D preserves semantic information, and is less severe in reducing bias from gendered words as it is on gender-neutral words. The difference in spread is less apparent in the original BERT model. We also note that debiasing the embeddings of BERT rather than the attention mechanism as in Kaneko & Bollegala (2021) (Figure 3(c)) is not enough since bias information is still lurking (and perhaps made worse for some words) in the attention component. Thus, we conclude that working on attention directly constitutes our best option for debiasing to date.

## A.6 EFFECT OF ATTENTION-BASED DEBIASING ON OTHER TRANSFORMER-BASED TEXT ENCODERS

We evaluate five widely used sentence-level text encoders: BERT (Devlin et al., 2018), ALBERT (Lan et al., 2019), RoBERTa (Liu et al., 2019), DistilBERT (SANH et al.) and SqueezeBERT (Iandola et al., 2020). For each model, we evaluate both its base and large variants (except for DistilBERT and SqueezeBERT since these are not available in HuggingFace's transformers library[15]), original and debiased; which gives a total of sixteen evaluated models. We use Crows-Pairs dataset (Nan-

---

[15]https://huggingface.co/transformers/index.html

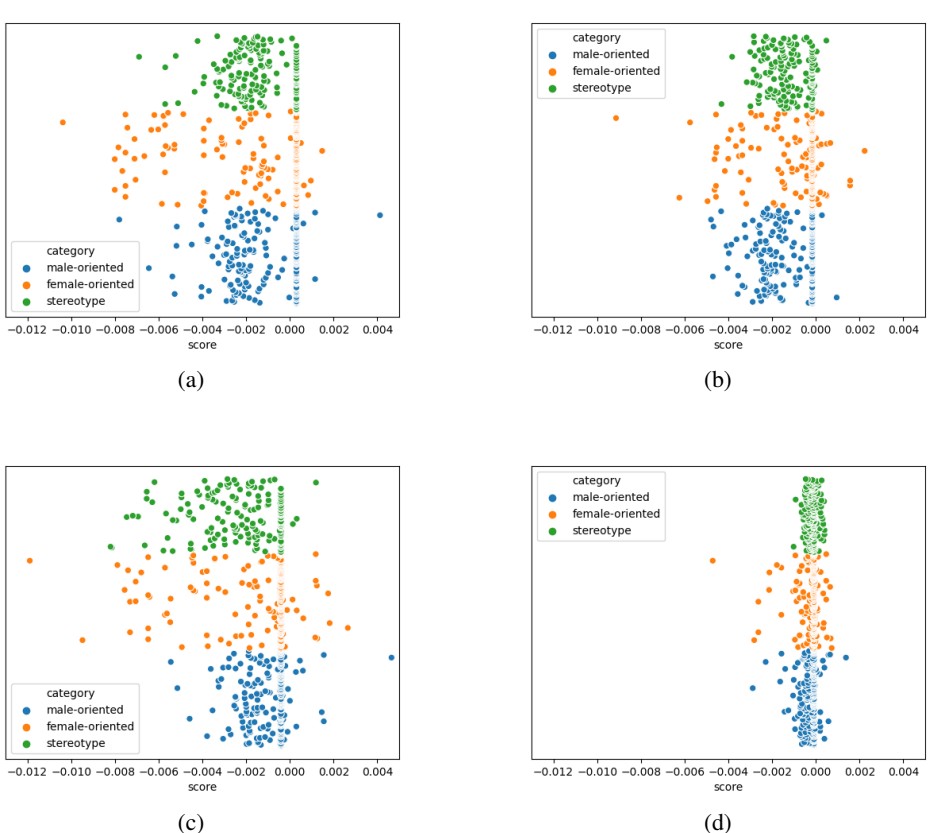

Figure 3: Scatter plots of attention scores on male - female direction. (a) Original BERT, (b) BERT debiased by Sent-D (c) BERT debiased by Kaneko & Bollegala (2021), (d) BERT debiased by Att-D

Table 9: Bias reduction in BERT base and large measured on Crows-Pairs dataset. Each cell is organized as follows: o → d +/-diff where $o$ is the stereotype score of the original model, $d$ is that of the debiased model using attention-based debiasing, and $diff$ is the difference in stereotype score. Negative values correspond to reduction in bias (desired) where positive values mean addition of bias (undesired).

| Models | BERT base | | BERT large | |
|---|---|---|---|---|
| Overall | 60.48 → 55.70 | -04.78 | 59.68 → 56.96 | -02.72 |
| race | 58.14 → 51.15 | -06.99 | 60.08 → 53.49 | -06.59 |
| gender | 58.02 → 57.36 | -00.66 | 55.34 → 53.05 | -02.29 |
| socioeconomic | 59.88 → 51.16 | -08.72 | 56.40 → 57.56 | +01.16 |
| nationality | 62.89 → 57.86 | -05.03 | 52.20 → 57.23 | +05.03 |
| religion | 71.43 → 64.76 | -06.67 | 68.57 → 66.67 | -01.90 |
| age | 55.17 → 43.68 | +01.15 | 55.17 → 54.02 | -01.15 |
| sexual orientation | 67.86 → 58.33 | -09.53 | 65.48 → 67.86 | +02.41 |
| physical appearance | 63.49 → 61.90 | -01.89 | 69.84 → 65.08 | -04.76 |
| disability | 61.67 → 60.00 | -01.67 | 76.67 → 65.00 | -11.67 |

gia et al.) to quantify the intensity of undesired stereotypes encoded therein. As a reminder, ideal stereotype scores according to Crows-Pairs benchmark should be close to 50, i.e. models preferring neither stereotypes nor anti-stereotypes. Tables 9, 10, 11 and 12 show the bias results for BERT, ALBERT, RoBERTa and DistillBERT/SqueezeBERT respectively.

All five models exhibit substantial levels of bias, and in each of the bias types with differing intensities (religion, sexual orientation and disability being the bias categories with the most severe stereotyping). Also, we find that the large variants are more biased than their base counterparts mainly because large models, with their larger capacity and greater number of parameters, can capture more intricate and more sophisticated aspects of training data, exposing them to learn more bias. This finding corresponds well to results of previous work (Nangia et al.; Nadeem et al., 2020). The tables also show that Att-D is effective in mitigating bias from BERT, ALBERT, RoBERTa, DistilBERT and SqueezeBERT, and produces a reduction of up to 25%. We note that Att-D succeeds in debiasing all models, with varying effectiveness across bias types. We also note that Att-D meets the best success with ALBERT as reductions are greater on this particular text encoder. We believe this is because ALBERT is composed of a single transformer layer (Lan et al., 2019) with substantially less parameters than BERT or RoBERTa; which makes debiasing easier since there is no interference between different attention layers. Finally, we see from the tables that Att-D sometimes contributes to adding a bit of bias. We observe that this phenomenon is rare, and happens especially with bias types we did not include in our design[16]. We assume that not explicitly compelling the text encoder to equalize attention heads corresponding to these overlooked bias types gave it green light to adjust these attentions in a way to facilitate solving the optimization problem; even if it entails adding bias. We plan to include all bias types present in Crows-Pairs dataset to our debiasing design as a future work.

## A.7 APPLICATION-ORIENTED BIAS EVALUATION

Recent studies show that intrinsic metrics of bias do not necessarily correlate with bias measures on concrete real-world applications (Goldfarb-Tarrant et al., 2020). In the body of this paper, we already conducted intrinsic and extrinsic bias evaluations. In this experiment, we validate the efficacy of our debiasing method on a concrete real-world hate speech detection application where an input snippet of text is classified as either offensive (*toxic*, *harmful*, *disrespectful*...) or not. We use hate speech detection because it is well studied in the literature (Burnap & Williams, 2016; Ribeiro et al., 2018; Zhang et al., 2018), and high-quality datasets which are tagged with social groups already exist (Borkan et al., 2019; Mathew et al., 2021).

---

[16]In the current version of this work, we remind that we only consider five bias types: gender, race, religion, age and sexual orientation

Table 10: Bias reduction in ALBERT base and large measured on Crows-Pairs dataset. Each cell is organized as follows: o → d +/-diff where $o$ is the stereotype score of the original model, $d$ is that of the debiased model using attention-based debiasing, and $diff$ is the difference in stereotype score. Negative values correspond to reduction in bias (desired) where positive values mean addition of bias (undesired).

| Models | ALBERT base | | ALBERT large | |
|---|---|---|---|---|
| Overall | 56.76 → 51.99 | -04.77 | 60.48 → 53.58 | -06.90 |
| race | 51.36 → 48.84 | -00.20 | 59.11 → 50.97 | -08.14 |
| gender | 54.20 → 53.44 | -00.76 | 56.11 → 48.47 | -04.58 |
| socioeconomic | 60.47 → 61.05 | +00.58 | 54.07 → 50.00 | -01.16 |
| nationality | 51.57 → 57.86 | +06.29 | 62.26 → 60.38 | -04.07 |
| religion | 59.05 → 60.00 | +00.95 | 76.19 → 61.90 | -14.29 |
| age | 65.52 → 42.53 | -08.05 | 54.02 → 54.02 | -00.00 |
| sexual orientation | 75.00 → 38.10 | -13.10 | 71.43 → 63.10 | -08.33 |
| physical appearance | 46.03 → 41.27 | +04.76 | 58.73 → 57.14 | -01.59 |
| disability | 86.67 → 61.67 | -25.00 | 73.33 → 58.33 | -15.00 |

Table 11: Bias reduction in RoBERTa base and large measured on Crows-Pairs dataset. Each cell is organized as follows: o → d +/-diff where $o$ is the stereotype score of the original model, $d$ is that of the debiased model using attention-based debiasing, and $diff$ is the difference in stereotype score. Negative values correspond to reduction in bias (desired) where positive values mean addition of bias (undesired).

| Models | RoBERTa base | | RoBERTa large | |
|---|---|---|---|---|
| Overall | 53.98 → 51.39 | -02.59 | 61.27 → 56.83 | -04.44 |
| race | 47.09 → 50.39 | -02.52 | 61.43 → 53.49 | -07.94 |
| gender | 54.96 → 45.80 | -00.76 | 51.91 → 51.91 | -00.00 |
| socioeconomic | 56.40 → 55.81 | -00.59 | 66.28 → 59.88 | -06.40 |
| nationality | 45.28 → 43.40 | +01.88 | 56.60 → 55.35 | -01.25 |
| religion | 56.19 → 60.00 | +03.81 | 59.05 → 62.86 | +03.81 |
| age | 64.37 → 56.32 | -08.05 | 71.26 → 62.07 | -09.19 |
| sexual orientation | 69.05 → 48.81 | -17.86 | 71.43 → 59.52 | -11.91 |
| physical appearance | 66.67 → 60.32 | -06.35 | 68.25 → 66.67 | -01.58 |
| disability | 71.67 → 65.00 | -06.67 | 66.67 → 70.00 | +03.33 |

Table 12: Bias reduction in DistilBERT and SqueezeBERT measured on Crows-Pairs dataset. Each cell is organized as follows: o → d +/-diff where $o$ is the stereotype score of the original model, $d$ is that of the debiased model using attention-based debiasing, and $diff$ is the difference in stereotype score. Negative values correspond to reduction in bias (desired) where positive values mean addition of bias (undesired).

| Models | DistilBERT | | SqueezeBERT | |
|---|---|---|---|---|
| Overall | 56.83 → 51.26 | -05.57 | 57.43 → 54.71 | -02.72 |
| race | 53.29 → 47.87 | -01.16 | 55.04 → 56.01 | +00.97 |
| gender | 54.58 → 46.56 | -01.14 | 52.67 → 48.47 | -01.14 |
| socioeconomic | 55.81 → 58.14 | +02.33 | 57.56 → 51.16 | -06.40 |
| nationality | 54.09 → 50.94 | -03.15 | 53.46 → 61.01 | +07.55 |
| religion | 70.48 → 57.14 | -13.34 | 74.29 → 60.95 | -13.34 |
| age | 59.77 → 48.28 | -08.05 | 55.17 → 48.28 | -03.45 |
| sexual orientation | 70.24 → 55.95 | -14.29 | 70.24 → 57.14 | -13.10 |
| physical appearance | 55.56 → 63.49 | +07.93 | 52.38 → 52.38 | -00.00 |
| disability | 61.67 → 56.67 | -05.00 | 70.00 → 61.67 | -08.33 |

Admittedly, common social biases have also been shown to exist in hate speech detection models, for example in associating toxicity to frequently attacked groups (such as "muslim" or "gay") even if the text itself is not toxic (Dixon et al., 2018; Park et al., 2018). In this experiment, we adopt the bias definition of Borkan et al. (2019) which casts bias as a skewing in the hate speech detector scores based solely on the social groups mentioned in the text. In other words, we consider a model to exhibit unintended social stereotypes if the model's performance varies across groups. We use the bias measures proposed by Borkan et al. (2019) which are based on the Area Under the Receiver Operating Characteristic Curve (ROC-AUC, or AUC) metric. AUC measures the probability that a randomly chosen negative example (not offensive) receives a lower toxicity score than a randomly chosen positive example (offensive), meaning that a perfect model should always have an AUC score of 1.0. Stated differently, all negative examples have lower toxicity scores than positive examples. While AUC is used to measured the general performance of classifiers, Borkan et al. (2019) propose three extensions of AUC to measure bias. We summarize them in the following:

**Subgroup (Sub) AUC:** where AUC is computed only on the group under consideration and not on all the examples of the test benchmark, i.e. only positive and negative examples of the target group are considered. This metric represents the model's performance on a given group. A higher value means that the model is good at distinguishing between toxic and non-toxic texts specific to the group.

**Background Positive Subgroup Negative (BPSN) AUC:** where AUC is calculated on the negative examples of the target group, and the positive examples of the background (all other groups except the group under consideration). This metric computes whether the model *discriminates* against the target group with respect to the others. This value is reduced when non-toxic examples of the group have *higher* toxicity scores than actually toxic examples of the background.

**Background Negative Subgroup Positive (BNSP) AUC:** where AUC is calculated on the positive examples of the target group, and the negative examples of the background. This metric computes whether the model *favors* the target group with respect to the others. This value is reduced when toxic examples of the group have *lower* toxicity scores than non-toxic examples of the background.

In this experiment, we finetune the text encoder under study on hate speech detection task using the training set of HateXplain dataset (Mathew et al., 2021). We also use the test portion of HateXplain for the evaluation, which contains posts from Twitter[17] and Gab[18] annotated with their ground-truth toxicity scores and the social groups and communities they target. Fundamentally, the three metrics described above give bias scores per group. In order to combine the per group scores in one overall measure, we apply the Generalized Mean of Bias (GMB) introduced by the Google Conversation AI Team as part of their Kaggle competition[19], and later used by Mathew et al. (2021) in their own evaluations. The formula of GMB is as the following:

$$GMB(b) = (\frac{1}{|b|} \sum_{g=1}^{|b|} b_g^p)^{1/p} \tag{4}$$

where $b$ is an array of AUC scores per group, and $b_g$ is the AUC score of group $g$. We follow Mathew et al. (2021) and set p to *-5*. We compute the GMB of all three metrics: Subgroup, BPSN and BNSP. As for Subgroup, we also add the standard deviation as it gives valuable information about how much the performance of the hate speech detection model varies across groups. We report our results in Table 13, in addition to classic performance measures.

We observe that Att-D provides competitive results across the four bias metrics, and largely outperforms the baselines. Especially with *GMB-BNSP*, where bias scores of the original model are very low (i.e. it is throttled by social biases), we observe the best improvements overall, and by a large margin compared to existing debiasing methods. Also, the variance in model performance is lowest with Att-D, which confirms that the corresponding hate speech detection model has less stereotypes about different social groups. Finally, the general performance (Accuracy, F1 score and AUC) of the hate speech detection model after debiasing is not hurt.

---

[17]https://twitter.com

[18]https://gab.com

[19]https://www.kaggle.com/c/jigsaw-unintended-bias-in-toxicity-classification/overview/evaluation

Table 13: AUC-based bias measures on hate speech detection task

| Models | Performance | | | Bias | | | |
|---|---|---|---|---|---|---|---|
| | Acc↑ | F1↑ | AUC↑ | STD-Sub↓ | GMB-Sub↑ | GMB-BPSN↑ | GMB-BNSP↑ |
| BERT | 0.783 | 0.823 | 0.870 | 0.119 | 0.698 | **0.800** | 0.379 |
| Sent-D | 0.791 | 0.825 | 0.870 | 0.121 | 0.689 | 0.725 | 0.583 |
| Kaneko | 0.797 | 0.833 | 0.872 | 0.112 | 0.705 | 0.789 | 0.512 |
| Att-D | 0.789 | 0.829 | 0.866 | **0.085** | **0.808** | 0.793 | **0.726** |

### A.8 WORD-LEVEL VS SENTENCE-LEVEL DEBIASING

As previously explained in the paper, Att-D calibrates the attention weighs of all tokens of the input sentence on group-related words. Since we used BERT-based models in our experiments, the first token in the input is the special [CLS] token, which is considered by the NLP community as a vector representation for the entire input sentence. In the current version of Att-D, we also calibrate the attention weighs of the special [CLS] token on groups, in addition to calibrating the other tokens of the sentence. One can see this notion as a combined word-level and sentence-level debiasing. In this experiment, we motivate this design choice by comparing it to word-level and sentence-level debiasing separately. For word-level, we exclude the [CLS] token from the attention equalization process, whereas in sentence-level we only calibrate the attention of [CLS]. We use all the bias evaluations run so far to understand the difference in performance. Tables 14, 15, 16, 17 and 18 report the results of StereoSet, Crows-Pairs, inference, hate speech and GLUE experiments respectively. We denote word-level debiasing by **No [CLS]**, and sentence-level debiasing by **Only [CLS]** in the tables. The combination of both is referred to as **Att-D**, and is the variant that we promote in this paper. We observe that while the three settings are good at reducing bias from text encoders, Att-D is superior than word-level and sentence-level debiasing since it capitalizes on the benefits of both. It enjoys the fine granularity of reducing bias from every word, while it also mitigates biases that manifest at sentence-level.

### A.9 STATIC VS RANDOM ORDERING OF GROUP-RELATED WORDS

In the preprocessing step of our method (as explained in Section 3.2.1), we use a preset ordering of group-related words of a given bias type to form the second input. For example, if we have the groups *Muslim*, *Christian*, *Jew* and *Buddhist* defining the religion bias type, Att-D constructs the second input using the same preset ordering of groups across all samples of the training data. Continuing the example above, Att-D appends the following artificial sentence "muslim, christian, jew, buddhist". In this experiment, we change the ordering of groups in a random way. Tables 14, 15, 16, 17 and 18 also report the bias scores of Att-D (static ordering) and Att-D with random ordering.

Although the semantic performance of Att-D with random ordering is better, we notice that it suffers from a stronger presence of bias than in its static counterpart. In Table 17, Att-D with random ordering has an AUC score of 0 in one of the groups, which made the GMB extremely small. We suspect that the relatively poor fairness of random ordering owes to the fact that the model might be confused by different orderings throughout the iterations. A more serious analysis of the impact of group order on the overall performance (fairness and semantics) of Att-D motivates the direction of future work.

Table 14: Language modeling (lm) and Stereotype scores (ss) on StereoSet of different variants of Att-D

| Models | Att-D | | No [CLS] | | Only [CLS] | | Random Order | |
|---|---|---|---|---|---|---|---|---|
| Overall (lm/ss) | 83.34 | **53.04** | 80.37 | 53.71 | 81.70 | 55.51 | 82.91 | 54.75 |
| gender (lm/ss) | 78.24 | 53.73 | 76.86 | **52.94** | 75.88 | 54.51 | 79.02 | 55.69 |
| race (lm/ss) | 86.28 | **51.87** | 84.10 | 53.01 | 85.24 | 55.09 | 86.75 | 54.57 |
| religion (lm/ss) | 88.46 | **53.85** | 84.62 | 60.26 | 85.26 | 56.41 | 87.18 | 56.41 |
| profession (lm/ss) | 80.96 | **54.14** | 76.63 | **54.14** | 78.99 | 56.24 | 79.17 | 54.51 |

Table 15: Bias measurements of different variants of Att-D on Crows-Pairs

| Models | Att-D | No [CLS] | Only [CLS] | Random Order |
|---|---|---|---|---|
| Overall | 55.7 | 56.1 | **55.5** | 58.36 |
| gender | 57.36 | 50.76 | **50.0** | 53.82 |
| race | **51.15** | 54.84 | 53.1 | 57.75 |
| religion | **64.76** | 69.52 | 65.71 | 67.62 |
| age | 43.68 | 56.32 | 44.83 | **54.02** |
| sexual orientation | **58.33** | 71.43 | 63.1 | 64.29 |
| nationality | 57.86 | **53.46** | 65.41 | 62.28 |
| disability | 60.0 | 61.67 | **58.33** | 65.0 |

Table 16: Inference-based bias measurements on different variants of Att-D. Best scores are highlighted with **bold character**, underlined, or marked with † for **gender**, race and religion† respectively

| Model | Bias type | NN | FN | $\tau$:0.5 | $\tau$:0.7 |
|---|---|---|---|---|---|
| Att-D | gender | 01.31 | 00.43 | 00.35 | 00.21 |
| | race | 93.31 | 93.94 | 93.90 | 93.04 |
| | religion | 68.51† | 69.08† | 68.95† | 66.97† |
| No [CLS] | gender | 00.85 | 00.36 | 00.30 | 00.20 |
| | race | 76.14 | 76.24 | 76.19 | 74.26 |
| | religion | 40.80 | 40.04 | 39.98 | 37.78 |
| Only [CLS] | gender | **02.35** | **01.60** | **01.38** | **00.90** |
| | race | 81.63 | 81.52 | 81.50 | 80.37 |
| | religion | 44.40 | 44.01 | 43.95 | 42.76 |
| Random Order | gender | 01.54 | 00.51 | 00.39 | 00.23 |
| | race | 54.71 | 54.92 | 54.89 | 52.49 |
| | religion | 26.94 | 26.67 | 26.59 | 24.58 |

Table 17: AUC-based bias measures on hate speech detection task on different variants of Att-D

| | Performance | | | Bias | | | |
|---|---|---|---|---|---|---|---|
| Models | Acc↑ | F1↑ | AUC↑ | STD-Sub↓ | GMB-Sub↑ | GMB-BPSN↑ | GMB-BNSP↑ |
| Att-D | 0.789 | 0.829 | 0.866 | **0.085** | **0.808** | 0.793 | **0.726** |
| No [CLS] | 0.791 | 0.830 | 0.871 | 0.114 | 0.710 | **0.797** | 0.530 |
| Only [CLS] | 0.765 | 0.805 | 0.838 | 0.142 | 0.660 | 0.766 | 0.636 |
| Random Order | 0.784 | 0.822 | 0.861 | / | / | 0.764 | / |

Table 18: GLUE performance of different variants of Att-D. The table shows *accuracy* scores for **sst2**, **rte**, **wnli**, and **mnli** for both matched and mismatched instances; *f1* for **mrpc**; *spearman correlation* for **stsb**; and *matthews correlation* for **cola**

| Models | Single sentence tasks | | Double sentence tasks | | | | |
|---|---|---|---|---|---|---|---|
| | sst2 | cola | stsb | mrpc | mnli (m/mm) | rte | wnli |
| Att-D | 92.66 | 55.22 | **89.62** | 91.22 | **84.63** / 84.19 | 70.40 | **53.52** |
| No [CLS] | 91.51 | 40.85 | 88.94 | 91.62 | 84.49 / 84.02 | 68.95 | 40.85 |
| Only [CLS] | 92.43 | 55.23 | 89.43 | 90.04 | 84.42 / 84.67 | **71.84** | 23.94 |
| Random Order | **93.23** | **59.07** | 88.85 | **91.94** | 83.75 / **84.86** | **71.84** | 30.99 |

