# OpenReview forum: "Debiasing Pretrained Text Encoders by Paying Attention to Paying Attention"
_ICLR.cc/2022/Conference — ICLR 2022 Submitted_

### Official Review · Reviewer_CGM2 · 2021-11-02

**Correctness:** 3
**Technical Novelty And Significance:** 3
**Empirical Novelty And Significance:** 3
**Recommendation:** 6
**Confidence:** 4

**Main Review:**

The method for bias mitigation introduced by the paper is intuitive
and relatively simple. In particular, I liked the idea of sample
augmentation as opposed to trying to identify parts of the original
text that refers to social groups and trying to alter the weights
attending to those tokens, which is much higher to achieve.

I have a couple of concerns:

1. The method requires a teacher, unaltered model to be trained in
parallel, which can be computationally involved. I would have liked to
understand how much the performance of the model (perhaps on the GLUE
tasks) is changed if such a teacher model is not used.

2. I am concerned about the evaluation of bias. As recent studies
showed, intrinsic measures of bias do not necessarily correlate with
bias measures on a concrete, down-stream task that can be used in an
application. https://arxiv.org/abs/2012.15859 The paper includes the
NLI task, however, I perceive that task very similar to the intrinsic
tasks. It doesn't provide insights in what it would happen wrt bias in
a down-stream task. I would have liked the evaluation of the model to
include a task for which group annotations exist and the bias for the
original and modified model are presented. For example, the Jigsaw
dataset could be used for such an evaluation
(https://www.kaggle.com/c/jigsaw-unintended-bias-in-toxicity-classification/data),
or, for a smaller dataset, HateXplain could be used
(https://github.com/hate-alert/HateXplain). Without such an evaluation
for an application oriented task, it is hard to understand whether the
few points gained by the new model will lead to significant bias
improvement in a downstream task.

You may want to take a look at this paper that lists some of the problems with benchmarks such as StereoSet:
https://www.microsoft.com/en-us/research/uploads/prod/2021/06/The_Salmon_paper.pdf

I am not sure if I understood the need to separate the original input
from the artificially constructed one. Do you think that the model
will get confused if no separated is used and, thus, its semantic
performance would decrease?


**Summary Of The Paper:**

This paper addresses potential biases introduced by attention models
by re-weighing the attention weights. First, the paper provides a few
examples that demonstrate attention weights correlated with social
stereotype (e.g., doctor attending to he and nurse attending to
she). Then, it proposes to reduce this type of bias by "calibrating"
the attention weights. To do so, each sample is augmented with pairs
of words corresponding to the groups for which the model is intended
to mitigate bias. The augmented samples are used in training, and,
thus, each token in the input will attend to tokens in the augmented
portion of the sample. The weights are changed such that the weights
attending on the augmented, group-relevant tokens are similar. This
type of re-weighing can lead to some of the semantic meaning encoded
in the network to change. To prevent this change, the attention
weights corresponding to the tokens from the original sample are
forced to follow the weights in an unaltered model that is trained in
parallel. Last, the paper suggested also utilizing negative sampling:
using random words for augmentation and forcing all attention weights
to follow the ones of the teacher.

For evaluation, the paper shows the performance of several
transformer-based models.  The evaluation of the method includes three
different benchmarks: two benchmarks that measure bias intrinsically
(Crows-Pairs and StereoSet) and an NLI task designed for measuring
bias. To make sure that the semantic strengths of the model are not
lost, the models are also evaluated using the GLUE tasks. The model
obtains competitive performance for the GLUE tasks, while reducing
bias compared to original models and a couple of related works.

**Summary Of The Review:**

The method presented in the paper is intuitive and simple, with some,
potentially considerable computation overhead. While the semantic
performance is preserved and there are modest improvements on
intrinsic measures of bias, the effects of the model on the bias
results of a downstream, application task are not addressed and,
hence, not understood.

---

> ### Author Response · Authors · 2021-11-21
> **Answer to Reviewer CGM2**
>
> Thank you for your positive and insightful comments. Thank you also for the time and effort spent reviewing our paper. We are glad to hear that you liked our approach. We also appreciate your evaluation suggestion. We are happy to announce that we conducted the evaluation on HateXplain, and the results are very encouraging. We added them to the revised version of the paper. For your concerns:
>
> **The method requires a teacher, unaltered model to be trained in parallel, which can be computationally involved**
>
> Although our method does require a teacher, the latter is not trained during debiasing. The teacher is the original text encoder which has been previously pre-trained on language modeling (usually) prior to debiasing. In this work, we use the teacher as a reference to the original attention weights. We only train the student (which is the model to debias), but we do not alter the teacher, since this defeats the purpose of using a teacher in the first place. So, we don’t believe that our method is that computationally involved, or at least not as much as the reviewer has first assumed.
>
>
> **I am concerned about the evaluation of bias**
>
> We thank the reviewer for the suggested reference. Indeed, we are aware that Crows-Pairs and StereoSet have been criticized. However, we believe that it is necessary to evaluate bias from different angles and draw a complete evaluative picture. In our opinion, evaluations of bias should include intrinsic, extrinsic and visual evaluations. Extrinsic evaluations (Section 4.2.2) and the visualizations in the appendix (Section A.5) demonstrate that our method is better than the baselines, and these evaluations are trustworthy to begin with. On the other hand, all intrinsic evaluations have been criticized to date [1], including SteroSet and Crows-Pairs. For completeness of the evaluation, we believe it is necessary to include at least one intrinsic measure. And as the authors of [2] surmise that Crows-Pairs and StereoSet are more stable than other intrinsic measures of bias, we used them in this paper. Besides, we thank the reviewer for the excellent suggestion of evaluating with HateXplain dataset on a real downstream task which is hate speech detection. We report the positive results of this experiment in the appendix of the revision (Section A.7).
>
>
> **I perceive [the NLI task] very similar to the intrinsic tasks**
>
> To the best of our understanding, the NLI task is not an intrinsic task, but an extrinsic one as it measures bias on a downstream inference task, which is a fundamental component in many NLP-based applications. This experiment shows that our method makes inferences less biased. That being said, we suspect that the reviewer would have liked to see evaluations where the metric is based on a variant of performance gap between different groups. We include such an experiment based on the reviewer suggestion (thanks!) in the revision using the HateXplain dataset (Section A.7). We Are happy to announce that the results are positive.
>
>
> **I am not sure if I understood the need to separate the original input from the artificially constructed one**
>
> We don’t want to pollute the original input sentences with group-related words because doing so damages the grammatical correctness of the sentence, and may change its meaning. Besides, finetuning on grammatically-bleached sentences might confuse the model and compel it to lose its pre-acquired notions of grammar and semantics. That is why we separate the original sentence from the artificially-constructed one.
>
> [[1] https://arxiv.org/pdf/2012.15859.pdf](https://arxiv.org/pdf/2012.15859.pdf)
>
> [[2] https://arxiv.org/pdf/2105.05641.pdf](https://arxiv.org/pdf/2105.05641.pdf)
>
> We hope that our answers and the revised paper are satisfying and that they would increase your confidence in recommending the acceptance of our paper. Please do not hesitate to let us know if there is any additional material and/or detail we can offer.

---

> > ### Comment · Reviewer_CGM2 · 2021-11-22
> > **Results on HateXplain**
> >
> > Glad to hear you found the suggestion useful and that the results for HateXplain task are promising. Good luck with your work!

---

### Official Review · Reviewer_7L6Q · 2021-11-03

**Correctness:** 2
**Technical Novelty And Significance:** 2
**Empirical Novelty And Significance:** 2
**Recommendation:** 3
**Confidence:** 3

**Main Review:**

Strength
- The debiasing based on attention is a new method.
- The empirical results seem good for likelihood based and inference based evaluation.

Weakness
- While the work has its merit in decreasing the biases in some metrics, It is bad using those dataset as leaderboard without looking into them. [1] elaborates on the few problems with CrowS-Pairs and StereoSet.
- In [2], it criticizes all the work focusing on word level debiasing and those are really not removing them, but hiding them. To me, this paper is no different from prior work in this aspect. Also given most of the differences are quite small, I am not sure whether the paper is making meaningful progress.
- The proposed approach is quite limited in the setup. How does it handle multiple token words, different synonyms? Also, as described in section 5, it is not scalable for words that need not be debiased.
- The proposed approach is also not very novel. It is an extension of counterfactural data augmentation in the attention with regularization with original attention.


[1] https://aclanthology.org/2021.acl-long.81/

[2] https://arxiv.org/abs/1903.03862



**Summary Of The Paper:**

This papers propose a attention-based debiasing method for pretrained text encoders. The paper shows reduction in bias on few datasets and a small degradation to the GLUE benchmark.


**Summary Of The Review:**

Overall, while there are merits in proposing attention based debiasing, the evaluation methodology is weak. It is another paper treating fairness issues as benchmarking without questioning on the actual effect, differences, and whether the datasets are meaningful in the first place.

---

> ### Author Response · Authors · 2021-11-21
> **Answer to Reviewer 7L6Q**
>
> We would like to thank you for your thoughtful comments, and the time and effort spent reviewing our paper. We hope that our answer sheds some light on the evaluation choices that we took, and clears the small misunderstanding on a couple of concerns that we found in the review. To address your concerns:
>
> **While the work has its merit in decreasing the biases in some metrics, It is bad using those dataset as leaderboard without looking into them. [Ref] elaborates on the few problems with CrowS-Pairs and StereoSet.**
>
> We thank the reviewer for the good point raised here and the paper suggestion. We are aware of the few problems found with Crows-Pairs and StereoSet. For more reliable experiments, we added other types of bias evaluations (inference-based experiment in Section 4.2.2 and the visualizations in the appendix A.5) which show that our method is good at reducing bias. We believe that evaluations of bias should be comprehensive and include at least one intrinsic evaluation, independent from any downstream task. We specifically chose  StereoSet and Crows-Pairs for this purpose because the authors of [1] surmise that they are more stable than other intrinsic measures of bias (WEAT and SEAT for example). We also add a new experiment in the revision (Section A.7) which was suggested by another reviewer, that quantifies bias on a real downstream application with crowd-sourced data. The new experiment is regarded as trustworthy by the community, and confirms that our method succeeds in reducing bias from the original text encoder. We also observe that our method outperforms existing baselines in all the experiments (both intrinsic and extrinsic).
>
> **In [Ref], it criticizes all the work focusing on word level debiasing and those are really not removing them, but hiding them. To me, this paper is no different from prior work in this aspect. Also given most of the differences are quite small, I am not sure whether the paper is making meaningful progress.**
>
> We believe that there has been a small misunderstanding regarding this aspect. In this work, we make each token of $s$ (which is the tokenized version of the input sentence) pay equal amounts of attention to the tokens of $s_g$ (group-related words). However, in most existing text encoders, the first token is the **[CLS]** token, which is considered by the community as a vector representation for the entire sentence. In this work, we also calibrate the attention of the **[CLS]** token on groups, in addition to calibrating the other tokens’ attention. In this spirit, our method becomes a mixture of word-level and sentence-level debiasing. We admit that the passage in the paper is prone to misunderstanding if the reader is not familiar with how BERT models function. We will of course clarify it in the revision. Prior to this paper submission, we have already compared our method with cases when we don’t use the **[CLS]** token (word-level debiasing) and when only the **[CLS]** token is calibrated (sentence-level debiasing), and found that using **[CLS]** and all the words is the best approach. We will report our findings in the appendix of the revision (Section A.8).
>
> **The proposed approach is quite limited in the setup. How does it handle multiple token words, different synonyms? Also, as described in section 5, it is not scalable for words that need not be debiased.**
>
> We take this opportunity to clarify why the limitations do exist but are not particularly significant:
> * Even though we did not consider multiple-word tokens in our experiments, extending our method beyond single-word tokens is relatively straightforward. Suppose we want to calibrate the attention of a word $w$ on two groups: group **A** (described with a three-word token $A_1$ $A_2$ $A_3$), and group **B** (described with a two-word token $B_1$ $B_2$). Debiasing in this case can be conducted by equalizing the attention of $w$ between the sum of its attention on $A_1$, $A_2$ and $A_3$, and the sum of its attention on $B_1$ and $B_2$, such that the final attention of $w$ on all words of **A** is equal to that of all words of **B**. The same mechanism can be easily adapted to non-binary bias types.
> * For synonyms, we can add them as different tuples of the same bias type.
> * As for the words that need not not be debiased, we mention in the conclusion that this is a limitation of our work. Compiling words that need not be debiased for every bias type can be very expensive to do. We mitigate this problem via knowledge distillation, where we force the attention of such words not to diverge too much from the original ones provided by the teacher.
>
> [[1] https://arxiv.org/pdf/2105.05641.pdf](https://arxiv.org/pdf/2105.05641.pdf)
>
> We answer the last concern in another slot due to space limitations...

---

> ### Author Response · Authors · 2021-11-21
> **Answer to Reviewer 7L6Q (Continuation)**
>
> Here, we a address the last concern of the reviewer:
>
> **The proposed approach is also not very novel. It is an extension of counterfactural data augmentation in the attention with regularization with original attention.**
>
> To the best of our knowledge, we are not aware of previous works that focus on debiasing at attention-level. We join the other two reviewers on the novel quality of our approach. We also believe that our method is not an extension of counterfactual Data Augmentation (CDA), at least not in the traditional meaning of the term. While CDA perturbs data in order to generate new training examples, we only append group-related tokens to the original data in order to make debiasing feasible, not to generate new examples. Although this notion holds its attraction and constitutes a sound and an interesting future direction for our research.
>
> We hope that our answers and the revised version of the paper are satisfying, and that they succeed in making the reviewer change his/her mind about the acceptance of the paper. If you need any more detail, please feel free to let us know.

---

### Official Review · Reviewer_KjSU · 2021-11-04

**Correctness:** 3
**Technical Novelty And Significance:** 3
**Empirical Novelty And Significance:** 3
**Recommendation:** 6
**Confidence:** 4

**Main Review:**

Strengths:
1. While we have seen a number of interesting papers focusing on debiasing static word embeddings, debiasing (even measuring) contextualized word embeddings is underexplored.  As pretrained text encoders become more and more powerful, this paper is approaching a very challenging and important problem.

2. The way that the proposed method to correct the attention scores is smart. More specifically, it attaches a sequence of words from different social groups to the original training sentence, which makes it very controllable to adjust the attention scores assigned to specific words.

3. The proposed method also has an advantage of reducing multiple types of biases (e.g. gender, race, age, ...) simultaneously and there can be more than 2 groups for certain biases (although in Table 1, examples are all binary).

Weakness:
1.  The experimental results, especially in Table 4, are not very convincing. The proposed method seems only work well on reducing race bias. Gender bias seems to be the hardest one to mitigate across all methods and Kaneko is doing better on reducing religion bias.
2. The paper also need to provide more details on hyper-parameter tuning, e.g. lambda in Eq (3). And also more details and ablation studies on negative sampling and layer selection.
3. I am not fully convinced about the motivation (intuition) in Sec. 3.1. Figure1 definitely presents some good examples of bias in encoders but it is hard to conclude that bias is mostly from the encoder. The paper needs more quantitative analysis on bias in encoders and also decoders to support this claim.
4. The paper mentioned that bias measurements like WEAT have been questioned. However, Stereo-set and Crows-Pairs have also been criticized [1].
5.I am wondering that the positions of words in s_g may also affect the attention scores. I may have missed something here but has any analysis on the relative positions has been done? E.g. "man" comes first or "woman" comes first. If not, it would be interesting to see if we need to randomly change the order of words in the tuples.

[1] Stereotyping Norwegian Salmon: An Inventory of Pitfalls in Fairness Benchmark Datasets


**Summary Of The Paper:**

This paper proposes a new debiasing method for contextualized word embeddings, specifically for attention-based text encoders. At a very high level, the proposed method tries to calibrate the attention scores of words from different groups, e.g. to reduce gender bias, the method forces the model (text encoder) to allocate same attentions to word "man" and "woman". Experimentally, the paper also demonstrates relatively good results on both likelihood-based evaluation (StereoSet and Crows-Pairs) and inference-based evaluation (NLI).

**Summary Of The Review:**

A paper with a novel method to reduce biases in text encoders and also reasonable empirical results. I would recommend to accept this paper.

---

> ### Author Response · Authors · 2021-11-21
> **Answer to Reviewer KjSU (Part 1)**
>
> We are glad to see your positive comments, and happy to hear that you find our approach smart and novel. As for your concerns:
>
> **The proposed method also has an advantage of reducing multiple types of biases (e.g. gender, race, age, ...) simultaneously and there can be more than 2 groups for certain biases (although in Table 1, examples are all binary).**
>
> In this work, we addressed five bias types (gender, race, religion, age and sexual orientation) of which only three (gender, age and sexual orientation) were set to be binary. The purpose of Table 1 is to illustrate what our group tuples look like, and we chose to show gender and age because of space limitations on the paper only. Including race and religion (four classes each) in the table as well would have consumed a lot of valuable space. The full list of our bias types and their corresponding tuples can be found in the appendix (Table 6).
>
>
> **The experimental results, especially in Table 4, are not very convincing. The proposed method seems only work well on reducing race bias. Gender bias seems to be the hardest one to mitigate across all methods and Kaneko is doing better on reducing religion bias.**
>
> We agree that in the NLI experiment, gender bias seems the hardest one to mitigate. For this experiment, we followed the evaluation setup of [1] and evaluated gender bias associated with occupations. For example, we may have the premise “The doctor bought a dress” or “The dancer bought a dress”, and as a hypothesis “The man bought a dress”. On the other hand, race and religion are associated with polarity. For example, we may have “The violent person prepared lunch” as a premise and “The muslim prepared lunch” as hypothesis. We suspect that gender bias was harder to mitigate because it may have been confused with occupation bias in the evaluation dataset. In the example above, if we have “The doctor bought a dress” as a premise, the model may already regard this sentence as confusing and contradictory with its latent knowledge, and lean toward predicting “Contradiction” rather than “Neutral” without even looking at the hypothesis. With this in mind, all baselines struggle with gender, but our method is still better in reducing this complex notion of gender bias than related works. Most importantly, bias is reduced from the original text encoder (comparison to the first three rows in Table 4). However, our method works well on race and religion as the reviewer points out, because the dataset used for these bias types is free from occupation bias mingling in between. Kaneko is slightly better than our method in religion, but we largely outperform them in gender and race by a large margin.
>
>
> **The paper also need to provide more details on hyper-parameter tuning, e.g. lambda in Eq (3). And also more details and ablation studies on negative sampling and layer selection.**
>
> We conducted hyperparameter tuning manually on a development set that is 20% of the published News-commentary-v15 corpus, to which we append group tuples as in the training set. The final values of our hyperparameters are already present in the appendix (Section A.1). The appendix also includes some evaluations and small ablation studies on layer selection (Section A.3) and negative sampling (Section A.4).
>
> **I am not fully convinced about the motivation (intuition) in Sec. 3.1. Figure1 definitely presents some good examples of bias in encoders but it is hard to conclude that bias is mostly from the encoder. The paper needs more quantitative analysis on bias in encoders and also decoders to support this claim.**
>
> We agree with the reviewer that bias can manifest in both the encoder and decoder parts of a transformer. However, we focus in this work on models based only on the encoder side such as BERT, ALBERT, RoBERTa, DistillBERT… which previous work have already established as biased. We build on those findings and do not conduct further experiments to quantify bias on such models. We also do not focus on the decoder side, for example models of the GPT family. The purpose of the example in Section 3.1 is to show that social biases are also encoded in the attention layer, complementary to the commonly-held belief that bias is encoded in the embeddings. Our work goes from there, attempts to reduce bias from the attention layer, and shows that bias is reduced overall. Additional evaluations and quantifications of bias at attention-level constitutes a promising direction for future research.
>
> We will continue our answers in a second slot...
>
> [[1] https://arxiv.org/pdf/1908.09369.pdf](https://arxiv.org/pdf/1908.09369.pdf)

---

> > ### Author Response · Authors · 2021-11-21
> > **Answer to Reviewer KjSU (Part 2)**
> >
> > Again, we are happy to see your positive comments and your recommendation for the paper's acceptance. Here, we expand on your last two concerns.
> >
> > **The paper mentioned that bias measurements like WEAT have been questioned. However, Stereo-set and Crows-Pairs have also been criticized**
> >
> > We thank the reviewer for the suggested paper. We will point out the limits of and criticism to Crows-Pairs and StereoSet. In the evaluations of bias, we attempt to include intrinsic, extrinsic and visual evaluations. Extrinsic evaluations (Section 4.2.2) and the visualizations in the appendix (Section A.5) demonstrate that the method we propose is better than the baselines, and these evaluations are trustworthy to begin with. On the other hand, all intrinsic evaluations have been criticized to date [2], including SteroSet and Crows-Pairs. For completeness of the evaluation, we include at least an intrinsic measure. Since the authors of [3] surmise that Crows-Pairs and StereoSet are more stable than other intrinsic measures of bias, we used them in this paper. As another reviewer suggested, we add a new experiment in the revision (Section A.7), that quantifies bias on a real downstream application with crowd-sourced data. The new experiment confirms that our method is better than the baselines, and succeeds in reducing bias from the original text encoder.
> >
> >
> > **I am wondering that the positions of words in s_g may also affect the attention scores. I may have missed something here but has any analysis on the relative positions has been done? E.g. "man" comes first or "woman" comes first. If not, it would be interesting to see if we need to randomly change the order of words in the tuples.**
> >
> > The reviewer raises a good point here. Indeed, attention scores do change if we change the order of words. We have already tried our method with a random ordering of group-related words, but found that a preset ordering is slightly better than a random one. We suspect that it may be less confusing for the model to encounter the same ordering during training. However, a deeper analysis of word ordering is an interesting research direction as we have previously stated in our conclusion. We will add the random ordering experiment in the appendix of the revision (Section A.9)
> >
> > [[1] https://arxiv.org/pdf/1908.09369.pdf](https://arxiv.org/pdf/1908.09369.pdf)
> >
> > [[2] https://arxiv.org/pdf/2012.15859.pdf](https://arxiv.org/pdf/2012.15859.pdf)
> >
> > [[3] https://arxiv.org/pdf/2105.05641.pdf](https://arxiv.org/pdf/2105.05641.pdf)
> >
> > We thank you for your time and effort spent reviewing our paper. We hope that our answers are satisfying and that they would increase your confidence in recommending the acceptance of the paper. Please do not hesitate to let us know if there is any additional material and/or detail we can offer.

---

### Author Response · Authors · 2021-11-22
**Revision summary**

We would like to thank all reviewers for spending time and effort in reviewing our work and providing valuable feedback. In our answers, we did our best to address all of your comments. We hope that our answers are satisfactory. Your suggestions can definitely help us to improve our work. Below please find a short summary of our revisions to the paper:

* Added a footnote to explain that we also use the **[CLS]** token in attention calibration, and that this mechanism enables our approach to be both word-level and sentence-level debiasing. This is to avoid the potential misunderstanding that R_7L6Q raised.
* Added an experiment that measures bias on a downstream hate speech detection task as R_CGM2 suggested, using the HateXplain dataset.
* Added an experiment that compares the performance of word-level, sentence-level, and the combination thereof in debiasing. This experiment shows that our approach is better than exclusive word-level or sentence-level debiasing. This experiment was done before the initial submission but was not included in the paper then. We add it in the appendix of the revised version.
* Added an experiment that compares the effect of a random ordering of words in contrast to our approach which uses a preset ordering of group-related words. We find that random ordering meets less success than a static ordering of words. This experiment was done before the initial submission but was not included in the paper then. We add it in the appendix of the revised version.

If you have any further comments about revisions before the end of the discussion period, please don't hesitate to let us know.

---

### Decision · Program_Chairs · 2022-01-20

**Decision:**

Reject

**Comment:**

This paper presents a debiasing technique that modifies a model's attention mechanism by equalizing attention across social groups.  The authors show that their approach (which is perhaps the first of its kind to look at transformer based models and debiasing instead of fixed word representations) work well in debiasing across certain social group indicators while maintaining overall performance.  However, there is disagreement between reviewers in terms of acceptance of the paper (especially Reviewer 7L6Q wants the paper to be rejected and points to recent critiques such as https://aclanthology.org/2021.acl-long.81.pdf that point out pitfalls with the benchmarks used in this paper).  I agree with said reviewer that a lot of these benchmarks are toy-ish and finding real impact of bias in NLP models is quite elusive.  Hence, I am recommending the paper be rejected for ICLR 2022 and the suggestions below be incorporated towards a better draft for the future.